# Plasma Nucleosomes in Primary Breast Cancer

**DOI:** 10.3390/cancers12092587

**Published:** 2020-09-10

**Authors:** Michal Mego, Katarina Kalavska, Marian Karaba, Gabriel Minarik, Juraj Benca, Tatiana Sedlackova, Paulina Gronesova, Dana Cholujova, Daniel Pindak, Jozef Mardiak, Peter Celec

**Affiliations:** 12nd Department of Oncology, Faculty of Medicine, Comenius University and National Cancer Institute, 83310 Bratislava, Slovakia; jmardiak@gmail.com; 2Translational Research Unit, Faculty of Medicine, Comenius University and National Cancer Institute, 83310 Bratislava, Slovakia; katarina.kalavska@nou.sk; 3Department of Oncosurgery, National Cancer Institute, 83310 Bratislava, Slovakia; marian.karaba@nou.sk (M.K.); juraj.benca@nou.sk (J.B.); daniel.pindak@nou.sk (D.P.); 4Institute of Molecular Biomedicine, Faculty of Medicine, Comenius University, 81372 Bratislava, Slovakia; gabriel.minarik@gmail.com (G.M.); tatiana.sedlackova@gmail.com (T.S.); petercelec@gmail.com (P.C.); 5Department of Medicine, St. Elizabeth University, 81102 Bratislava, Slovakia; 6Biomedical Center, Slovak Academy of Sciences, 84505 Bratislava, Slovakia; paulina.gronesova@gmail.com (P.G.); dana.cholujova@savba.sk (D.C.); 7Department of Oncosurgery, Slovak Medical University, 83101 Bratislava, Slovakia

**Keywords:** primary breast cancer, circulating nucleosomes, circulating tumor cells, plasminogen activator inhibitor-1, cytokines

## Abstract

**Simple Summary:**

Nucleosomes composed of DNA and histone proteins enter the extracellular space and end eventually in the circulation when cells die. In blood plasma, they could represent a nonspecific marker of cell death, potentially useful for noninvasive monitoring of cancer. The aim of this study was to analyze circulating nucleosomes in relation to patient/tumor characteristics and prognosis in nonmetastatic breast cancer. This study included 92 patients with breast cancer treated with surgery. Plasma nucleosomes were detected in samples taken in the morning on the day of surgery. Circulating nucleosomes were positively associated with the systemic inflammation but not with other patient/tumor characteristics. Patients with lower nucleosomes had lower risk of disease recurrence compared to patients with higher nucleosomes. Our data suggest that plasma nucleosomes in nonmetastatic breast cancer are associated with systemic inflammation and might have a prognostic value. The underlying mechanisms require further studies.

**Abstract:**

When cells die, nucleosomes composed of DNA and histone proteins enter the extracellular space and end eventually in the circulation. In plasma, they might serve as a nonspecific marker of cell death, potentially useful for noninvasive monitoring of tumor dynamics. The aim of this study was to analyze circulating nucleosomes in relation to patient/tumor characteristics and prognosis in primary breast cancer. This study included 92 patients with breast cancer treated with surgery for whom plasma isolated was available in the biobank. Plasma nucleosomes were detected in samples taken in the morning on the day of surgery using Cell Death Detection ELISA kit with anti-histone and anti-DNA antibodies. Circulating nucleosomes were positively associated with the systemic inflammatory index (SII), but not with other patient/tumor characteristics. Patients with high SII in comparison to low SII had higher circulating nucleosomes (by 59%, *p* = 0.02). Nucleosomes correlated with plasma plasminogen activator inhibitor-1, IL-15, IL-16, IL-18, and hepatocyte growth factor. Patients with lower nucleosomes had significantly better disease-free survival (HR = 0.46, *p* = 0.05). In a multivariate analysis, nucleosomes, hormone receptor status, HER2 status, lymph node involvement, and tumor grade were independent predictors of disease-free survival. Our data suggest that plasma nucleosomes in primary breast cancer are associated with systemic inflammation and might have a prognostic value. The underlying mechanisms require further studies.

## 1. Introduction

Breast cancer is the most common diagnosed cancer and the leading cause of cancer death among women in developed countries [1]. Despite advances in cancer prevention, diagnoses, and treatment, still approximately 5% of patients are diagnosed with metastatic disease, and 20–30% of initially primary breast cancer develops metastasis subsequently, during the course of the disease.

Extracellular DNA (ecDNA), also called cell-free DNA, is present in blood plasma in various forms [2]. EcDNA in the circulation of cancer patients contains tumor DNA from the primary tumor, metastasis, or circulating tumor cells, as well as healthy host cells mostly of hematopoietic origin [3,4,5]. Plasma ecDNA is partially free unbound DNA and, so, sensitive to rapid cleavage, but it also can be protected as ecDNA hidden in apoptotic bodies and/or bound to proteins such as histones in the form of nucleosomes [5].

Nucleosomes are composed of DNA wound around histone proteins and represent the basic structural unit of chromatin in the nucleus [6]. After cell death, membranes and nuclei disintegrate and cell-free nucleosomes can get into the circulation. Plasma nucleosomes might serve as a nonspecific biomarker of cell death [7]. This might be of interest in patients not only with autoimmune diseases, but also with sepsis or cancer [8,9,10]. The prognostic value of the concentration of circulating nucleosomes was shown in several types of cancer including lung, pancreatic, or colorectal cancer [11,12,13,14,15]. For example, in pancreatic cancer, high nucleosome levels during treatment, but not pretherapeutic levels, correlate with time to progression [16]. Similarly, in non-small cell lung cancer, high baseline nucleosome level and/or during chemotherapy was associated with poor response to treatment and these data suggested that circulating nucleosomes are a valuable tool for early prediction of chemotherapy efficacy in cancer patients [17]. However, when it comes to primary breast cancer, data in the published literature are limited.

In this study, we aimed to analyze circulating nucleosomes in relation to patients/tumor characteristics and prognosis in primary breast cancer.

## 2. Methods

### 2.1. Study Patients

This study included 92 primary breast cancer patients (stage I–III) treated with surgery from March to November 2012, for whom plasma isolated in the morning on the day of surgery was available in the biobank. This study represents a substudy of a translational trial that aimed to evaluate prognostic value of circulating tumor cells in primary breast cancer [18]. Study eligibility criteria and study details were described previously [18]. The study was approved by the Institutional Review Board (IRB) of the National Cancer Institute of Slovakia (TRUSK002, 20.6.2011). Each participant provided signed informed consent before study enrollment.

### 2.2. Detection of Circulating Tumor Cells (CTCs) in Peripheral Blood

CTCs were detected in peripheral blood by a quantitative real-time polymerase chain reaction (qRT-PCR)-based assay of peripheral blood as described previously [18,19,20].

### 2.3. Plasma Isolation

Venous peripheral blood samples were collected in EDTA-treated tubes in the morning on the day of surgery and centrifuged at 1000× *g* for 10 min at room temperature within 2 h of venipuncture and processed as described previously [21].

### 2.4. Quantification of Circulating Nucleosomes

The commercially available Cell Death Detection kit (Roche, Basel, Switzerland) was used for the measurement of nucleosomes. Briefly, 20 mL of plasma was mixed with biotin-labeled anti-histone and peroxidase-conjugated anti-DNA antibodies. After incubation and washing, the substrate for the peroxidase enzyme was added. Absorbance was measured at 405 nm in arbitrary units after stopping the reaction. Interassay and intra-assay coefficients of variation were below 10% and 5%, respectively.

### 2.5. Measurement of DD, TF, uPA, and PAI-1 in Plasma

Plasma tissue factor (TF), d-dimer (DD), urokinase plasminogen activator (uPA), and plasminogen activator inhibitor-1 (PAI-1) were analyzed using enzyme-linked immunosorbent assays (ELISA) as described previously [21].

### 2.6. Plasma Cytokines and Angiogenic Factors Analysis

Plasma samples were analyzed for 51 plasma cytokines and angiogenic factors: TGF-β1, TGF-β2, TGF-β3, IFN-α2, IL-1α, IL-2Rα, IL-3, IL-12p40, IL-16, IL-18, CTACK, Gro-α, HGF, LIF, MCP-3, M-CSF, MIF, MIG, β-NGF, SCF, SCGF-β, SDF-1α, TNF-β, TRAIL, IL-1β, Il-1RA, IL-2, IL-4, IL-5, IL-6, IL-7, IL-8, IL-9, IL-10, IL-12, IL-13, IL-15, IL-17, Eotaxin, FGF basic, G-CSF, GM-CSF, IFN-γ, IP-10, MCP-1, MIP-1α, MIP-1β, PDGF bb, RANTES, TNF-α, VEGF using predesigned panels as described previously and were available for subset of patients (Bio-Plex Pro TGF-β assay, Bio-Plex Pro Human Cytokine 21- and 27-plex immunoassays; Bio-Rad Laboratories, Hercules, CA, USA) [22]. The large panel of cytokines was analyzed as data were available from the previous study [22].

### 2.7. Complete Blood Count and Inflammation-Based Scores

Complete blood count (CBC) and CBC-derived inflammation-based scores were calculated as described previously [23,24]. For CBC-derived inflammation-based scores, identical cut-off values as published previously for metastatic breast cancer patients were used [23,24]. Data for calculation of NLR, PLR, MLR, SII were available for 54, 52, 48, and 52 patients, respectively.

### 2.8. Statistical Analysis

The characteristics of patients is summarized using mean (range) for continuous variables and frequency (percentage) for categorical variables. The median follow-up period was calculated as the median observation time among all patients and among those who were still alive at the time of their last follow-up. Disease-free survival (DFS) was calculated from the date of blood sampling to the date of disease recurrence (locoregional or distant), secondary cancer, death, or last follow-up. DFS was estimated using the Kaplan–Meier product limit method and compared between groups by log-rank test. For survival analysis, circulating nucleosomes were dichotomized to “low” or “high” (nucleosome level below vs. above mean, respectively). Univariate analyses with Chi squared or Fisher’s exact test were performed to find associations between prognostic factors.

A multivariate Cox proportional hazards model for DFS was used to assess differences in outcome on the basis of the nucleosomes status (above mean vs. below mean), hormone receptor status (positive for either vs. negative for both), HER-2 status (positive or negative), axillary lymph node involvement (N0 vs. N+), grade (grade 3 vs. grade 1 and 2). Stepwise regression techniques were used to build multivariate models using a significance level of 0.10 to remain in the model. All p values presented are two-sided, and associations were considered significant if the p value was less than or equal to 0.05. Statistical analyses were performed using NCSS 11 Statistical Software (2016, NCSS, LLC., Kaysville, UT, USA, ncss.com/software/ncss).

## 3. Results

### 3.1. Patients’ Characteristics

The study population consisted of 92 primary breast cancer patients with a median age of 60 years (range: 25–83 years). The patient characteristics are shown in Table 1. There were 79 (85.9%) patients with estrogen receptor-positive (ER) and/or progesterone receptor-positive (PR) tumors, and 16 (17.4%) patients with HER2/neu-positive tumors.

### 3.2. Association between Nucleosomes and Patient/Tumor Characteristics

The characteristics of patients and the associations with circulating nucleosomes are shown in Table 2. The concentration of circulating nucleosomes was not associated with any patient/tumor characteristics except the systemic inflammatory index (SII), where patients with high SII had significantly higher levels of circulating nucleosomes compared to patients with low SII (0.17 vs. 0.27, *p* = 0.02). There was also a trend for higher level of circulating nucleosomes in patients with high neutrophil/lymphocyte ratio (*p* = 0.07). There was no association between molecular subtype and plasma nucleosomes, even if molecular subtypes of breast cancer were further segregated by tumor grade. We also analyzed association of chronic medication/comorbidities (Appendix A, Table A1) and circulating nucleosomes, but we found no association.

### 3.3. Association between Nucleosomes and Plasma Cytokines

Patients with nucleosomes above mean in peripheral blood had significantly elevated plasma IL-16 (*p* = 0.005), IL-18 (*p* = 0.0004), and hepatocyte growth factor (*p* = 0.043), as compared to patients with nucleosomes below mean, while there was an inverse correlation between nucleosomes and IL-15 (*p* = 0.036). There was also a trend for higher IFN-α2 (*p* = 0.055) and RANTES (*p* = 0.053) in patients with higher nucleosome level (Table 3).

### 3.4. Nucleosomes and Coagulation

There was no association between circulating nucleosomes and DD, TF, and/or uPA, while patients with nucleosomes above mean had significantly elevated levels of plasma PAI-1 (Table 4).

### 3.5. Prognostic Value of Nucleosomes on Disease-Free Survival in Primary Breast Cancer

At a median follow-up time of 55.0 months (range = 4.9–76.7 months), 23 patients (25.0%) had experienced a DFS event, and 15 patients (16.3%) had died. Herein, we present DFS analysis due to the immaturity of overall survival data. Patients with lower than mean nucleosomes had significantly better disease-free survival (HR = 0.46, 95% CI 0.19–1.12, *p* = 0.05) (Figure 1). The prognostic value of circulating nucleosomes was most pronounced in lymph node-positive disease with high proliferation rate and in patients with detectable circulating tumor cells with epithelial-to-mesenchymal transition, but negative for epithelial circulating tumor cells (Table 5). In a multivariate analysis, nucleosomes, hormone receptor status, HER2 status, lymph node involvement, and tumor grade were independent predictors of disease-free survival (Table 6).

Circulating nucleosomes added prognostic value also to prognostic value of CTC_EMT, where double-positive patients (positive for both CTC_EMT and high-circulating nucleosomes) had worse prognosis compared to all other groups of patients (Figure 2).

## 4. Discussion

In this translational study, circulating nucleosomes showed neither an association with basic patient/tumor characteristics nor a correlation to CTCs. The origin of circulating nucleosomes is unclear and likely complex [25]. While there is no correlation between CTCs and SII and/or neutrophil/lymphocyte ratio [23,24], this study showed for the first time an association between plasma nucleosomes and SII. Patients with high SII had significantly higher level of nucleosomes. Similarly, there was a trend of higher nucleosomes in patients with high neutrophil/lymphocyte ratio, however, the neutrophil/lymphocyte ratio is part of the SII.

Tumor-induced systemic changes in immune cells contribute to cancer progression and metastasis. Various forms of ecDNA including extracellular nucleosomes and naked ecDNA differ in their cytotoxic and proinflammatory effects [26]. For example, histones in the nucleosomes induce proinflammatory signaling via toll-like receptors (TLR2/4), with subsequent production of TNF-α, IL-6, IL-10, and myeloperoxidase, but they exhibit TLR-independent cytotoxicity as well [26,27,28]. On the other hand, the ecDNA as part of the nucleosomes is recognized by the TLR9 [29]. In our study, nucleosomes were associated with several proinflammatory cytokines, suggesting the association of circulating nucleosomes with systemic inflammation. Histones in the nucleosomes could induce formation of neutrophil extracellular traps (NETs), which contain nucleosomes and stimulate further NETs production in a positive feedback loop [27]. On the other hand, nucleosomes could induce different inflammatory pathways, as they, in contrast to histones, seem not to be cytotoxic to the endothelium [28]. The analyzed nucleosomes could be from tumor cells, but also from the released NETs. This would explain the observed association between circulating nucleosomes and systemic inflammation in primary breast cancer patients. NETs contain nuclear DNA and proteins that possess antibacterial characteristics crucial for fighting pathogens [30,31]. The same NETs, however, also induce intravascular coagulation [32] and their overproduction can lead to autoimmune diseases [33]. While circulating ecDNA correlates with activation of coagulation [34], we for the first time describe this association for circulating nucleosomes. Further research is needed to uncover if nucleosomes directly activate PAI-1, or if high PAI-1 is a marker of coagulation activation in more aggressive disease that leads to release of more nucleosomes.

Data on the prognostic value of plasma nucleosomes in breast cancer is limited. In a small study, nucleosomes were elevated in locally confined and metastatic breast cancer in comparison to healthy individuals. During neoadjuvant chemotherapy, patients with no change of a local disease had significantly higher pretherapeutic concentrations of nucleosomes than patients in remission [14]. In another study, plasma nucleosomes were higher in primary breast cancer patients when compared to healthy controls, and similarly to our study, there was no association between nucleosomes and patient/tumor characteristics [15]. Circulating nucleosomes were, however, not able to discriminate between benign and malignant breast lesions [35]. Their concentration was found to be associated with lymph node-positive breast cancer and the presence of distant metastases [35].

In our study, we observed an inferior outcome of primary breast cancer patients with high plasma nucleosomes. This is in contrast to a previous study, where elevated plasma nucleosomes were associated with a better prognosis in both node-negative and node-positive early breast cancer [15]. However, the nucleosome detection method as well as the cut-off value to discriminate “low” and “high” plasma nucleosomes was different compared to our trial and therefore, these differences in results could be due to these factors. In our trial, the prognostic value of nucleosomes was consistent in various subgroups, however, it was most pronounced in poor prognostic subgroups such as lymph node-positive disease with high proliferation rate and in patients with detectable circulating tumor cells with epithelial-to-mesenchymal transition. The prognostic value of circulating nucleosomes was independent from established prognostic markers and was confirmed in a multivariate analysis. Moreover, when we combined two circulating biomarkers, circulating tumor cells, and circulating nucleosomes, we were able to uncover a subgroup of patients with extremely poor prognosis with two-year DFS of only 33.3%.

Our study has some limitations. The major one is small sample size, especially for associations between inflammatory indexes and nucleosomes. This is associated with decreased statistical power of analyses and increased confidence intervals of results. Other limitations represent the data availability for analysis of association between circulating nucleosomes and various clinic–pathological parameters, which further decreases statistical robustness and could have an impact on study results. Circulating plasma nucleosomes increase in non-neoplastic disease processes including inflammation, autoimmune diseases, sepsis, and stroke. When we analyzed association between chronic medication/comorbidities and circulating nucleosomes, no association was found, however, none of our patients received anti-inflammatory drugs and/or had inflammatory disease that could affect study results. Another limitation is lack of follow-up analysis on patient samples collected postsurgery to examine whether the presurgery baseline levels of circulating plasma nucleosomes were altered postsurgery and whether this alteration in circulating nucleosome levels is correlated with decrease in systemic inflammatory index.

## 5. Conclusions

In conclusion, in this translational study, we have shown for the first time that circulating nucleosomes are associated with systemic inflammation and activation of coagulation in primary breast cancer. More importantly, we proved their prognostic value. While it is clear that the underlying mechanisms of nucleosome release, their origin, and their fate require further studies, we suggest that the quantification of plasma nucleosomes could be added to the established prognostic markers in breast cancer. Future trials should focus on validation of these results to establish prognostic utility of plasma circulation nucleosomes in addition to established prognostic factors.

## Figures and Tables

**Figure 1 cancers-12-02587-f001:**
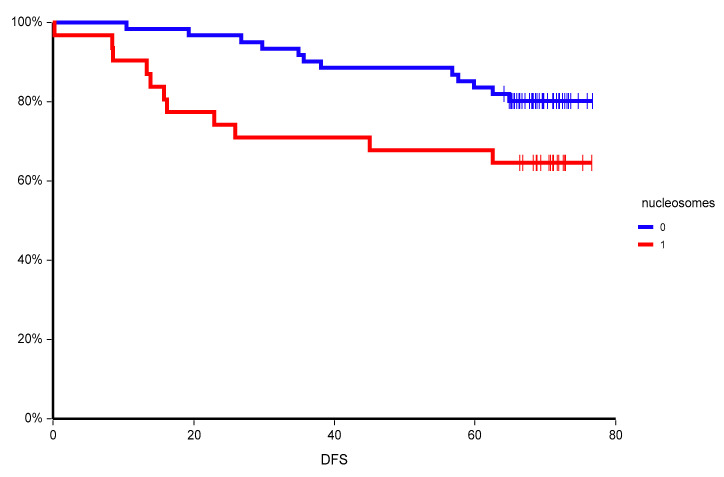
Kaplan–Meier estimates of probabilities of disease-free survival according to plasma nucleosome status in primary breast cancer patients (*n* = 92). HR = 0.46. 95% CI 0.19–1.12, *p* = 0.05, 0—nucleosomes below mean, 1—nucleosomes above mean.

**Figure 2 cancers-12-02587-f002:**
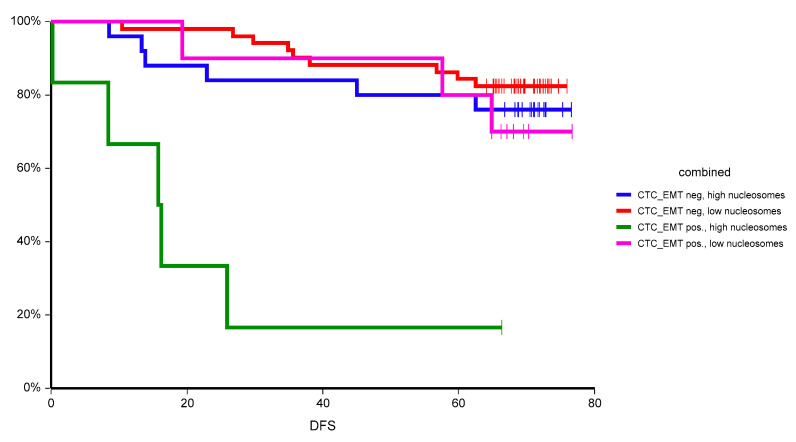
Kaplan–Meier estimates of probabilities of disease-free survival according to plasma nucleosome status and CTC_EMT in primary breast cancer patients (*n* = 92). Patients positive for CTC_EMT and high level of circulating nucleosomes had significantly worse survival compared to all other groups (*p* = 0.0000003).

**Table 1 cancers-12-02587-t001:** Patients’ characteristics.

Variable	N	%
**All Patients**	92	100.0
**T-stage**		
T1	58	63.0
>T1	34	37.0
**Histology**		
IDC	76	82.6
other	16	17.4
**Grade**		
low and intermediate	49	53.3
high grade	41	44.6
unknown	2	2.2
**Lymph nodes**		
N0	57	62.0
N+	34	37.0
unknown	1	1.1
**LVI**		
present	69	75.0
absent	23	25.0
**Hormone receptor status (cut-off 1%)**		
negative for both	13	14.1
positive for either	79	85.9
**Estrogen receptor-positive (cut-off 1%)**		
negative	16	17.4
positive	76	82.6
**Progesterone receptor-positive (cut-off 1%)**		
negative	25	27.2
positive	67	72.8
**HER2 status**		
positive	16	17.4
negative	76	82.6
**P53 status**		
negative	59	64.1
positive	32	34.8
unknown	1	1.1
**BCL-2**		
negative	27	29.3
positive	65	70.7
**unknown**		
**Ki67 status (cut-off 14%)**		
<14%	48	52.2
>14%	44	47.8
**Molecular subtype**		
Luminal A	43	46.7
Luminal B	36	39.1
HER2+	1	1.1
Triple-negative (TN)	12	13.0
**CTC EP**		
negative	75	81.5
positive	17	18.5
**CTC EMT**		
negative	76	82.6
positive	16	17.4
**CTC ANY**		
negative	62	67.4
positive	30	32.6

Abbreviations: CTC EP, circulating tumor cells with epithelial phenotype; CTC EMT, circulating tumor cells with epithelial–mesenchymal transition phenotype; CTC ANY, circulating tumor cells irrespective of phenotype; LVI, lymphovascular invasion.

**Table 2 cancers-12-02587-t002:** Association between nucleosomes and patient/tumor characteristics.

Variable	N	Mean	SEM	Median	*p*-Value
**All**	92	0.18	0.02	0.13	NA
**T-stage**					
T1	58	0.20	0.02	0.14	0.30
>T1	34	0.15	0.03	0.13	
**Histology**					
invasive ductal carcinoma	76	0.19	0.02	0.13	0.61
other	16	0.15	0.04	0.13	
**Grade**					
low and intermediate	49	0.20	0.03	0.14	0.91
high grade	41	0.16	0.03	0.13	
unknown	2				
**Lymph nodes**					
N0	57	0.18	0.02	0.12	0.10
N+	34	0.19	0.03	0.17	
unknown	1				
**Lymphovascular invasion**					
absent	69	0.18	0.02	0.13	0.20
present	23	0.19	0.04	0.16	
**Hormone receptor status (cut-off 1%)**					
negative for both	13	0.12	0.05	0.10	0.17
positive for either	79	0.19	0.02	0.14	
**HER2 status**					
negative	76	0.19	0.02	0.13	0.91
positive	16	0.17	0.04	0.15	
**P53 status**					
negative	59	0.19	0.02	0.14	0.51
positive	32	0.17	0.03	0.13	
unknown	1				
**BCL-2**					
negative	27	0.15	0.03	0.13	0.52
positive	65	0.20	0.02	0.13	
**Ki67 status (cut-off 14%)**					
<14%	48	0.20	0.03	0.15	0.38
>14%	44	0.16	0.03	0.13	
unknown					
**Molecular subtype**					
Luminal A	43	0.21	0.03	0.15	0.22
Luminal B	36	0.17	0.03	0.14	
HER2+	1	0.25	0.17	0.25	
Triple-negative (TN)	12	0.10	0.05	0.09	
**CTC EP**					
negative	75	0.18	0.02	0.13	0.44
positive	17	0.17	0.04	0.14	
**CTC EMT**					
negative	76	0.19	0.02	0.13	0.78
positive	16	0.16	0.04	0.13	
**CTC ANY**					
negative	62	0.19	0.02	0.13	0.19
positive	30	0.18	0.03	0.15	
**NLR (neutrophil/lymphocyte ratio) ***					
<3	43	0.18	0.03	0.12	0.07
>3	11	0.26	0.06	0.17	
**PLR (platelet/lymphocyte ratio) ***					
<210	43	0.19	0.03	0.12	0.71
>210	9	0.21	0.07	0.15	
**MLR (monocyte/lymphocyte ratio) ***					
<0.34	40	0.20	0.03	0.12	0.60
>0.34	8	0.15	0.07	0.14	
**SII (systemic inflammatory index) ***					
<836	40	0.17	0.03	0.10	**0.02**
>836	12	0.27	0.06	0.17	

Abbreviations: CTC EP, circulating tumor cells with epithelial phenotype; CTC EMT, circulating tumor cells with epithelial–mesenchymal transition phenotype; CTC ANY, circulating tumor cells irrespective of phenotype. * Data for calculation of NLR, PLR, MLR, SII were available for 54, 52, 48, and 52 patients, respectively; NA, not applicable. *p*-Values < 0.05 are written in Bold.

**Table 3 cancers-12-02587-t003:** Association between nucleosomes and plasma cytokines.

Variable	N	Mean	SEM	Median	*p*-Value
**IFN_a2 (ng/mL)**					
nucleosomes low	57	101.6	3.2	102.2	0.055
nucleosomes high	26	114.5	4.7	114.7	
**IL_16 (ng/mL)**					
nucleosomes low	58	349.2	19.8	330.9	**0.005**
nucleosomes high	27	446.7	29.0	419.5	
**IL_18 (ng/mL)**					
nucleosomes low	57	60.4	12.9	33.8	**0.0004**
nucleosomes high	26	120.0	19.1	69.7	
**HGF (ng/mL)**					
nucleosomes low	58	760.8	184.7	222.7	**0.043**
nucleosomes high	27	1312.2	270.8	438.6	
**M_CSF (ng/mL)**					
nucleosomes low	58	10.7	1.5	6.9	0.066
nucleosomes high	27	14.8	2.2	12.4	
**IL_15 (ng/mL)**					
nucleosomes low	39	22.0	2.1	16.7	**0.036**
nucleosomes high	22	13.6	2.9	12.8	
**RANTES (ng/mL)**					
nucleosomes low	58	8890.2	816.2	7644.8	0.053
nucleosomes high	27	7185.2	1196.2	4352.3	

Abbreviations: SEM, standard error of the mean. *p*-Values < 0.05 are written in Bold.

**Table 4 cancers-12-02587-t004:** Association between nucleosomes and coagulation.

Variable	N	Mean	SEM	Median	*p*-Value
**Tissue factor (pg/mL)**					
nucleosomes low	61	66.2	2.2	60.4	0.464
nucleosomes high	31	62.1	3.0	60.0	
**D-dimer (ng/mL)**					
nucleosomes low	61	412.3	53.6	312.2	0.394
nucleosomes high	31	552.3	75.2	401.7	
**uPA (ng/mL) ***					
nucleosomes low	59	4.8	0.5	3.8	0.925
nucleosomes high	31	4.6	0.6	3.7	
**PAI_1 (pg/mL) ***					
nucleosomes low	59	285.5	21.6	269.2	**0.042**
nucleosomes high	31	387.4	29.8	305.2	

Abbreviations: SEM, standard error of the mean; uPA, urokinase plasminogen activator; PAI-1, plasminogen activator inhibitor-1. * uPA and PAI-1 were not determined in two patients. *p*-Values < 0.05 are written in Bold.

**Table 5 cancers-12-02587-t005:** Prognostic value of nucleosomes on disease-free survival in primary breast cancer (nucleosomes dichotomized below vs. above mean).

Variable	HR	95% CI Low	95% CI High	*p*-Value
**All**	0.46	0.19	1.12	**0.05**
**T-stage**				
T1	0.29	0.08	1.02	**0.04**
>T1	0.56	0.14	2.19	0.33
**Histology**				
IDC	0.35	0.14	0.91	**0.01**
other	0	0	0	0.33
**Grade**				
low and intermediate	0.31	0.07	1.34	0.09
high grade	0.48	0.15	1.55	0.15
**Lymph nodes**				
N0	0.86	0.15	4.92	0.86
N+	0.36	0.13	1.04	**0.03**
**Lymphovascular invasion**				
absent	0.46	0.14	1.53	0.15
present	0.54	0.15	1.97	0.31
**Hormone receptor status (cut-off 1%)**				
negative for both	0.36	0.04	3.33	0.21
positive for either	0.41	0.15	1.15	0.06
**HER2 status**				
negative	0.55	0.19	1.6	0.23
positive	0.3	0.06	1.55	0.09
**P53 status**				
negative	0.48	0.16	1.44	0.13
positive	0.39	0.08	1.86	0.20
**BCL-2**				
negative	0.25	0.05	1.17	**0.02**
positive	0.59	0.19	1.85	0.33
**Ki67 status (cut-off 14%)**				
<14%	0.73	0.11	4.71	0.72
>14%	0.35	0.12	1	**0.02**
**CTC EP**				
negative	0.31	0.12	0.83	**0.01**
positive	0	0	0	0.18
**CTC EMT**				
negative	0.68	0.23	2.03	0.46
positive	0.17	0.03	0.9	**0.01**
**CTC ANY**				
negative	0.45	0.13	1.52	0.14
positive	0.5	0.14	1.87	0.27

*p*-Values < 0.05 are written in Bold.

**Table 6 cancers-12-02587-t006:** Multivariate analysis of factors associated with disease-free survival.

Variable	HR	95% CI Low	95% CI High	*p*-Value
**Nucleosomes**				
above mean vs. below mean	2.67	1.12	6.36	**0.0268**
**Hormone receptor status (cut-off 1%)**				
positive for either vs. negative for both	0.30	0.11	0.80	**0.0164**
**HER2 status**				
amplified vs. nonamplified	3.06	1.21	7.79	**0.0187**
**Lymph nodes**				
positive vs. negative	6.56	2.50	17.21	**0.0001**
**Grade**				
grade 3 vs. grade 1 and 2	2.85	1.14	7.08	**0.0246**

*p*-Values < 0.05 are written in Bold.

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
