# Peer review of "Plasma Nucleosomes in Primary Breast Cancer"

_cancers, 2020, doi:10.3390/cancers12092587_

Round 1

Reviewer 1 Report

The authors have responded to queries fully and to a good degree, but I feel that all of these responses need incorporating into the manuscript. For example the reference by De Giorgi et al in response to how the cut-off values were made and the BMC Cancer reference 2016 in response to why the large panel of factors was included. 

Once all of these responses have been incorporated I feel that the paper is publishable.

Author Response

Thank you very much for this point. Both references are included in Method section (line 169 – reference 22, and line 174, reference 23)

Reviewer 2 Report

The objective of this study was to analyze the circulating nucleosomes levels in relation to characteristics of tumors in primary breast cancer patient samples. The authors have reported that circulating plasma nucleosomes associated with systemic inflammatory index (SII) but not with other tumor characteristics in primary breast cancer patients samples. The authors have reported that the levels of circulating nucleosome levels is high in patients with high systemic inflammatory index compared to patients with lower systemic inflammatory index. The authors have also reported patients with lower nucleosomes had significantly better DFS.   The authors have addressed most of the comments however some of the major concerns were not addressed in the revised manuscript. The manuscript could not be considered for publication in the current form as addressing these concerns would be pivotal to support the authors conclusions.   1) The authors implicate circulating nucleosomes have prognostic value and reported that patients with lower nucleosomes have significantly better DFS. However the authors could not perform follow-up analysis on patient samples collected post surgery to examine whether the pre-surgery baseline levels of circulating plasma nucleosomes were altered post surgery and whether this alteration in circulating nucleosome levels is correlated with decrease in systemic inflammatory index? The authors have mentioned about lack of the samples however it is very important experiment as it would allow the authors to demonstrate the association between circulating nucleosome, systemic inflammatory index and DFS. The current data shown by authors is only an indication that too based on very small sample size (52 samples; if we consider all the variables it would be even smaller) (in fact several studies have already shown increased circulating nucleosome levels in breast cancer) about association between nucleosomes and SII and data from follow-up experiments (as listed above) is vital to prove the association.    2) The authors conclusions about association of nucleosomes with NLR, PLR, MLR, and SII may require more sample size as the findings were based on very small sample size of 54, 52, 48, and 52 respectively for these experiments. There are different molecular subtypes of breast cancer and the associations reported by the authors were different between patients with high systemic inflammatory index and low systemic inflammatory index. If we factor in all these variables along with subtypes of breast cancer, the conclusions drawn by the authors are from even smaller sample size. The authors are requested to consider of increasing the sample size.      

Author Response

Thank you very much for these points. We agree with the reviewer, however, as these data are not available, we added this limitation to discussion. However, we recognize that this is a major study limitation and comprise its possibility for publishing.

Reviewer 3 Report

The authors, in the present work, attempt to determine the impact of circulating nucleosomes in disease progression of patients with breast cancer and correlate with clinicopathological parameters. While the idea is interesting, there are some concerns in relevance to methodological and statistical aspects of the presented work.

Minor Concerns

  1. Line 63: it is stated "plasma isolated on the day before surgery was available in the biobank", in contrast in line 73 it is stated "blood samples were collected in EDTA-treated tubes in the morning on the day of surgery". Specify the time of blood collection.  
  2. Line 69: give briefly more methodological protocol information for CTC detection
  3. How you measured and which is the cut off value for low and high nucleosome concentration?

Major Concerns

  1. Of the most important findings presented in this study are the correlation of nucleosomes with SII and DFS. However, both correlations are based on a very small cohort of BrCa patients. The correlation of nucleosome with SII was based on 52 cases (as generated from Table2) and with DFS on 92 patients showing a marginal statistical significance (p=0.05). Increasing the number of patients could give a clearer result (negative or positive) for the correlation of nucleosomes with DFS.
  2. The kit employed by the authors is a cell death detection kit. Cell death is a highly ubiquitous procedure in the human body during numerous normal and pathological procedures. In that way, free nucleosomes can be produced with different ratios during sleep and daytime, during exercise and any kind of trauma (e.g a bruise), during a simple infection, in relevance to age, in relevance to stress and many kinds of medication. With that in mind authors should quantify in control subjects (adult “healthy” women) the fluctuations levels of nucleosomes concentration in their blood. As their study population (92 patients) is very small and their findings could be related to chance.    

Author Response

The authors, in the present work, attempt to determine the impact of circulating nucleosomes in disease progression of patients with breast cancer and correlate with clinicopathological parameters. While the idea is interesting, there are some concerns in relevance to methodological and statistical aspects of the presented work.

 Minor Concerns

  1. Line 63: it is stated "plasma isolated on the day before surgery was available in the biobank", in contrast in line 73 it is stated "blood samples were collected in EDTA-treated tubes in the morning on the day of surgery". Specify the time of blood collection.  
  2. Line 69: give briefly more methodological protocol information for CTC detection

Thank you very much for this point. The corrected this discrepancy. Correct statement is “in the morning on the day of surgery”

  1. How you measured and which is the cut off value for low and high nucleosome concentration?

 Thank you very much for this point. Nucleosomes were measured as described in the Methods section “Quantification of circulating nucleosomes”, dichotomization was based on mean level, we added this information to the statistical section (in addition to Table 5 and Figure 1.

Major Concerns

  1. Of the most important findings presented in this study are the correlation of nucleosomes with SII and DFS. However, both correlations are based on a very small cohort of BrCa patients. The correlation of nucleosome with SII was based on 52 cases (as generated from Table2) and with DFS on 92 patients showing a marginal statistical significance (p=0.05). Increasing the number of patients could give a clearer result (negative or positive) for the correlation of nucleosomes with DFS.

Thank you very much for this point. We agree with the reviewer, however, as these data are not available, we added this limitation to discussion. However, we recognize that this is a major study limitation and comprise its possibility for publishing.

  1. The kit employed by the authors is a cell death detection kit. Cell death is a highly ubiquitous procedure in the human body during numerous normal and pathological procedures. In that way, free nucleosomes can be produced with different ratios during sleep and daytime, during exercise and any kind of trauma (e.g a bruise), during a simple infection, in relevance to age, in relevance to stress and many kinds of medication. With that in mind authors should quantify in control subjects (adult “healthy” women) the fluctuations levels of nucleosomes concentration in their blood.

Thank you very much for these points.

We agree with the reviewer, that chronic medication/co-morbidities could affect the study results. We added data of chronic medication in the last 6 months, that potentially could affect study results. No inflammatory, autoimmune diseases, sepsis, and stroke were noted in last 6 months. There were no differences in nucleosome levels based on chronic medication/co-morbidities.

We agree with the reviewer, that there could be circadian fluctuation of circulating nucleosomes. In our study all samples were collected in the morning on the day of surgery, therefore, analysis of adult healthy women as control group probably doesn´t add more to the analysis and/or interpretation.

  1. As their study population (92 patients) is very small and their findings could be related to chance.    

Thank you very much for this point. In this point, we disagree with statement that due to small sample size the significant results could be just due to chance, as this is in contradiction to the statistical meaning of p-value. Decreased statistical power could lead to false negative but not to false positive results.

Round 2

Reviewer 2 Report

The authors have addressed majority of the concerns however authors were not able to include data for couple of major concerns and have provided details on these limitations in the discussion section of the manuscript.

Reviewer 3 Report

the authors addressed most of the minor comments. The paper have significant limitations which are referred to the discussion.